# Multiscale based nonlinear dynamics analysis of heart rate variability signals

**Syed Zaki Hassan Kazmi**[1], **Nazneen Habib**[2], **Rabia Riaz**[1], **Sanam Shahla Rizvi**[3], **Syed Ali Abbas**[1], **Tae-Sun Chung**[4]*

**1** Department of Computer Science & Information Technology, University of Azad Jammu and Kashmir, Muzaffarabad, Pakistan, **2** Department of Sociology & Rural Development, University of Azad Jammu and Kashmir, Muzaffarabad, Pakistan, **3** Raptor Interactive (Pty) Ltd, Centurion, South Africa, **4** Department of Software, Ajou University, Suwon, South Korea

* tschung@ajou.ac.kr

**Data Availability Statement:** The data used in the study is publicly available at https://physionet.org/about/database/. We have used following specific datasets. • MIT-BIH Normal Sinus Rhythm Database. https://physionet.org/content/nsrdb/1.0.

## Abstract

Acceleration change index (ACI) is a fast and easy to understand heart rate variability (HRV) analysis approach used for assessing cardiac autonomic control of the nervous systems. The cardiac autonomic control of the nervous system is an example of highly integrated systems operating at multiple time scales. Traditional single scale based ACI did not take into account multiple time scales and has limited capability to classify normal and pathological subjects. In this study, a novel approach multiscale ACI (MACI) is proposed by incorporating multiple time scales for improving the classification ability of ACI. We evaluated the performance of MACI for classifying, normal sinus rhythm (NSR), congestive heart failure (CHF) and atrial fibrillation subjects. The findings reveal that MACI provided better classification between healthy and pathological subjects compared to ACI. We also compared MACI with other scale-based techniques such as multiscale entropy, multiscale permutation entropy (MPE), multiscale normalized corrected Shannon entropy (MNCSE) and multiscale permutation entropy (IMPE). The preliminary results show that MACI values are more stable and reliable than IMPE and MNCSE. The results show that MACI based features lead to higher classification accuracy.

## Introduction

Biological signals are output of complex integrated subsystem, whose behaviour evolves with time [1]. Due to the interaction of numerous subsystems, the biological systems have the capability to evolve and adjust their self in a dynamic environment. The hierarchy of structural sub- systems and coupling between them depict the biological systems operate across multiple spatial and temporal scales [2–4]. Hence, information extracted at single time scale may not be dynamically correct. Multiscale Entropy (MSE) was proposed by Costa et al. [2], for extracting information from biological signals and validated that biological signals provide dynamically incorrect information at single time scale. Since then, MSE has been applied in numerous fields including biomedical signal processing [2, 3, 5, 6], electro-seismic time series data [7] and financial time series [8]. Various variants of original MSE have been proposed by either

0/ • Normal Sinus Rhythm RR Interval Database. https://physionet.org/content/nsr2db/1.0.0/ • BIDMC Congestive Heart Failure Database. https://physionet.org/content/chfdb/1.0.0/ • Congestive Heart Failure RR Interval Database. https://physionet.org/content/chf2db/1.0.0/ • MIT-BIH Atrial Fibrillation Database. https://physionet.org/content/afdb/1.0.0/.

**Funding:** Basic Science Research provided support for this study through the National Research Foundation of Korea (NRF) funded by the Ministry of Education in the form of a grant awarded to TSC (2019R1F1A1058548). This funder played a role in study design, decision to publish, and preparation of the manuscript, but did not play a role in data collection or analysis. Raptor Interactive (Pty) Ltd. provided support in the form of a salary for SSR. The specific roles of this author are articulated in the 'author contributions' section. This funder played a major role in study design, data collection and analysis, decision to publish, or preparation of the manuscript.

**Competing interests:** The authors have read the journal's policy and have the following competing interests: SSR is an employee of Raptor Interactive (Pty) Ltd. This does not alter our adherence to PLOS ONE policies on sharing data and materials. There are no patents, products in development or marketed products associated with this research to declare.

changing the coarse graining procedure or by using different entropy estimate to quantify the dynamical information and to address its drawbacks [6, 9–12].

In MSE, initially a coarse-grained procedure is applied on the original time series, according to the scale factor τ. On scale 1, the resultant coarse-grained time series is the original time series itself. At scale two, the average of ith and (i+1)th value in the time series is taken to form the coarse-grained time series and its size is calculated by dividing the original time series with τ. Then, Sample Entropy (SE) of coarse-grained time series is computed and plotted in contrast to scale factor.

MSE required long size time series and the size of error bars increases in small time series, because size of the coarse-grained time series heavily depends on the scale factor. Original author of MSE used 20000 data points and calculated entropy for 20 scales. Similarly, MSE is also affected by non-stationarity, because of fixed size of similarity criterion (r). MSE was successfully applied for healthy (NSR) and diseased (CHF) subjects. Diseased subject had lower complexity as compared to healthy subjects [2, 3].

MSE uses sample entropy on the coarse-graining time series, whereas Multiscale Permutation Entropy (MPE) uses concept of Permutation Entropy (PE) on multiple scales of coarse-grained time series. PE is advantageous in the presence of dynamical noise, so it can be successfully used to characterize the healthy and diseased subjects [13]. Gaussian white noise was used to compute the MPE. The results of the computation were compared with Multiscale entropy (MSE). It was found that both SE and PE decreased monotonically as the scale factor increases [13]. Healthy (NSR) and diseased (CHF) subjects were compared and it was found that healthy subjects are more irregular on large time scales as compared to the original time series.

In MSE, increase in the scale factor results in a decreased coarse-grained time series, and the variance of the entropy's coarse-grained series is increased, which is estimated by SE. The variance would become high at large scale of estimated entropy values, which results in a decreased reliability, making time series difficult to distinguish. Composite Multiscale entropy (CMSE) was proposed to reduce the variance at large scales of estimated entropy, which showed good performance for short time series [10, 11]. Both 1/f noise and while noise data was used for computation of CMSE and MSE and while comparing the results of both estimates, it was found that CMSE gave reliable estimation of entropy than MSE.

Recently sign time series analysis methods were applied for quantifying the information carried by biological signals [14]. Ashkenazy et al. (2000) in their scaling analysis of RR time series, used the sign series (sign of difference of the time series) as an intermediary time series [14]. They derived two sub series: magnitude $m_i = |\Delta RR_i|$ and sign series $s_i = sign(\Delta RR_i)$ from the original time series. They obtained scaling exponent by using detrended fluctuation analysis (DFA) [15] on sign series. To quantify the dynamics information from inter beat interval time series data (healthy and pathological) Garcia-Gonzalez proposed Acceleration change index (ACI) [16]. ACI is closely related to the time series autocorrelation function. The study revealed that ACI was lower in control groups compared to pathological subjects ACI is robust in case of dynamical and observational noise, however, its classification ability is modest. Thus, using ACI for quantifying the dynamical information from coarse grained time series generated using multiscale coarse graining procedure can be an important endeavour for distinguishing healthy and pathological. In this study, we propose multiscale ACI (MACI) to investigate the dynamical fluctuations of interbeat interval time series for assessing cardiac autonomic control at multiple time scale improve classification ability of ACI for distinguishing healthy and pathological Subjects.

## Materials and methods

The scale-based acceleration change index named as multiscale acceleration change index (MACI) is the modified form of acceleration change index (ACI). ACI was proposed Garcia-

Gonzalez et al [16] for characterizing the dynamics of HRV signals [13], which is closely related to autocorrelation function of a given signal. Garcia-Gonzalez et al [16] described in detail the mathematical background relation between autocorrelation function and ACI of the time series. The results reported by Garcia-Gonzalez et al [16] aver that ACI was lower in control groups compared to post-infarct (PI) groups. On comparing with standard linear HRV measures such Mean RR, standard deviation of normal to normal RR-intervals (SDNN), root mean square of the successive differences (RMSSD) and ratio of low frequency to high frequency (LF/HF) with linear ACI was only index that provided significant no other indices except ACI provide significant difference between the groups [16].

The cardiac autonomic control maintained by balancing action of sympathetic and parasympathetic branches of the autonomic nervous system comprises of feedback controlling mechanism operating at multiple time scales. Traditional ACI did not consider multiple time scales and can have limited capability to classify healthy and different pathological groups. To address this issue, a novel approach MACI is proposed by incorporating multiple time scales to extract dynamical information encoded by the HRV signals about healthy and pathological systems.

## Acceleration change index (ACI)

To calculate ACI of time series data following procedure was adopted.

**Step 1:** Given the RR-interval time series. The differentiated RR-interval (DRR) (DRR) time series is obtained as:

$$DRR(n) = RR(n+1) - RR(n), n \in [N-1]$$

Where N is the total number of RR-intervals and RR(n) is the RR interval from beat n to beat n+1, and.

**Step 2:** Sign of DRR (SDRR) is obtained by quantizing DRR series in '0' and '1'. SDRR is '0' if DRR<0 and '1' if DRR> = 0.

**Step 3:** Starting from n = 1 to n = N—1, we generate the sign change (SC) series as

$$SC(j) = n$$

$$if \ \ SDRR(n) \neq SDRR(n-1), \ \ j \in [1, M+1]$$

Where M + 1 is the number of sign changes. SC(j) = n implies that RR(n) is the jth local maximum or minimum of the tachogram with respects to threshold T.

**Step 4:** Sign change series is differentiated to obtain DSC, which is the distance (in beats) between successive changes of sign of the DRR time series:

$$DSC(j) = SC(j+1) - SC(j)$$

The Acceleration change index (ACI) is defined as:

$$ACI = k/M$$

Where k is the number of times DSC time series equal 1 and M is the total number of samples of DSC time series.

## Multiscale acceleration change index (MACI)

To compute the corresponding ACI over a sequence of scale factors MACI is used. The coarse-grained time series, $y(\tau)$, can be constructed at a scale factor of $\tau$, for a one-dimensional

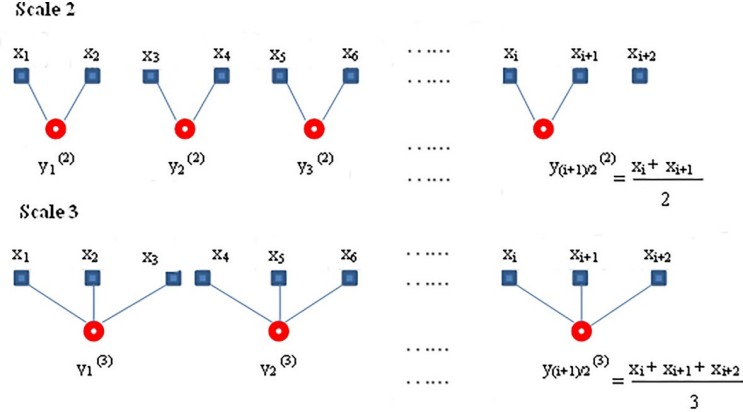

**Fig 1. Schematic illustration of the coarse grained procedure.**

time series, x = {x1,x2,. . .., xN}, according to the following equation:

$$y_j^\tau = \frac{1}{\tau}\sum_{i=(j-1)\tau+1}^{j\tau}x_i, \qquad 1 \le j \le \frac{N}{\tau}$$

The coarse-grained time series is divided into non-overlapping windows of length τ, and the data points inside each window are averaged as shown in Fig 1 [17, 18]. To find MACI value, we then define the ACI measurement of each coarse-grained time series.

$$MACI(x, \tau) = ACI(y^{(\tau)})$$

The MACI for each scale is computed by applying ACI on coarse-grained time series, $y^{(\tau)}$

The length of time series data is important for most of the nonlinear measurements. In MACI the length of time series after coarse-grained procedure is equal to the length of original time series divided by scale factor (τ). As the length of coarse-grained time series is reduced the variance of ACI measurements grows. Flow chart of MACI algorithm is presented in Fig 2.

## Statistical analysis and evaluation metrics

The analysis of variance (ANOVA) was used to analyse the differences among groups means and their concomitant measures for three or more groups. The one-way ANOVA is an omnibus approach, significant F statistic indicates that group means are statistically different, if any one of the groups is significantly different, but does not make paired comparison. The Bonferroni post-hoc test was used for multiple comparisons among the groups. The degree of separation among groups was quantified by using Area under receiver operator characteristic Curve (AUC) values (Mcneil & Hanley, 1984). The AUC is generally distinguished index for quantifying the level of separation between the groups. The AUC values show perfect separation of two groups at maximum value that is 1 and 0.5 value of AUC correspond to separation of groups by pure chance.

## Data sets

In this study we used data sets of normal sinus rhythm, congestive heart failure and atrial fibrillation. The details of these data sets are illustrated in Table 1.

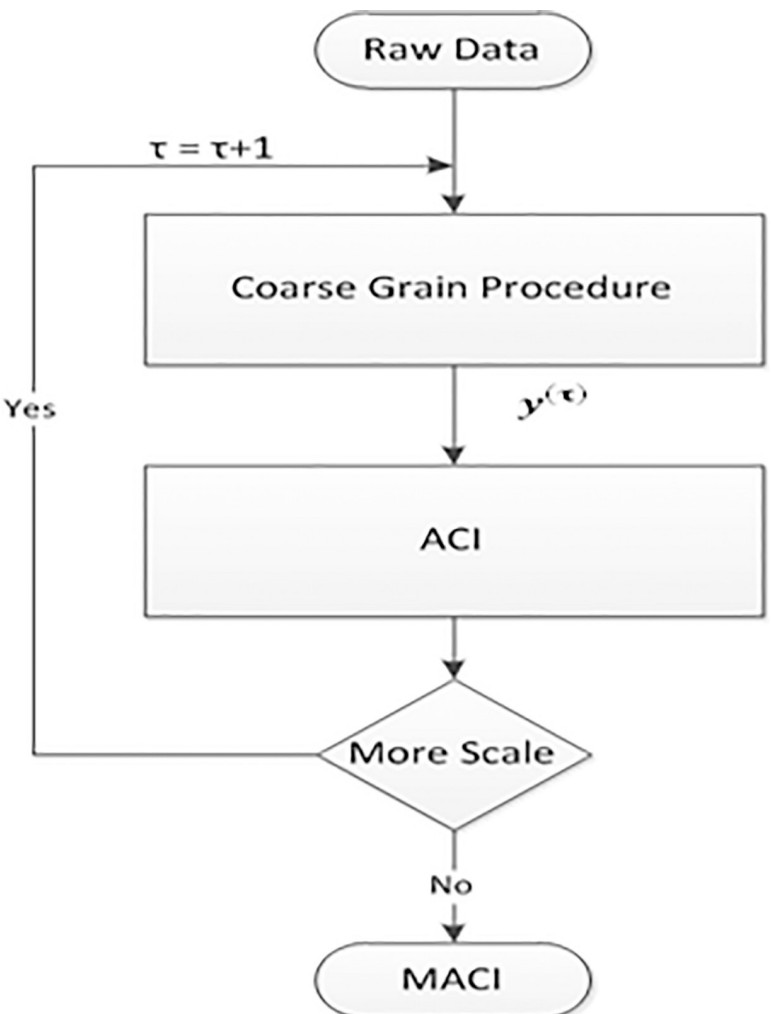

**Fig 2. Flow chart of the MACI algorithm.**

## Results

The performance of MACI was evaluated for distinguishing healthy and pathological groups at multiple time scale using coarse graining procedure. The mean of the time series was used as criterion for transforming original time series into coarse grained time series at multiple time scales. The results of MACI were compared with original ACI algorithm (MACI at temporal scale 1) for distinguishing healthy and pathological groups.

In the Table 2, results of ANOVA for comparing NSR, CHF and AF subjects using ACI and MACI(Mean) indices is presented. In the table columns are labelled Sum of Squares (between and within the groups), df (degrees of freedom between and within the groups), Mean Square (between and within the groups), F measure and Sig. The only column labelled Sig., representing significance level of ANOVA, is important from researcher's perspective for interpretation results and all other columns are used mainly for intermediary computational tasks. If the value (s) of Sig. in this column is (are) less than the critical value set by the researcher (say, 0.05), then the outcome will result in significant effects, while value(s) greater 0.05 value will depict that effects are not statistically significant. The statistically significant results

**Table 1. Data sets (R-wave to R-wave interval) used in the study.**

| Number of subjects | Data Sets | Source of Data |
|---|---|---|
| 72 | Normal sinus rhythm (NSR) | The 72 normal sinus rhythm subjects in which 35 male and 37 females were taken from publicly available databases of physio net. The age of the subjects was between 20 to 78 years. The 24 subjects taken from MIT-BIH NSR database comprised of 18 hours ECG recordings. Rest of 48 subjects taken from NSR RR interval database comprising of 54 beat annotation files of long-term ECG recording [19, 20]. |
| 44 | Congestive heart failure (CHF) | The 44 congestive heart failure subjects in which 29 male and 15 females were taken from physio net (publicly available database). The 15 sever CHF subjects were taken from BIDMC CHF database. Rest of 29 subjects taken from CHF RR interval database comprising of 29 beat annotation files for long term ECG recordings [20]. |
| 24 | Atrial fibrillation (AF) | The 24 atrial fibrillation subjects were taken from physio net a publicly available database. All-time series AF data sets comprise of 10 hours single recording [20]. |

demonstrate that differences between groups are not by chance or sampling error. The F-measure is basically the ratio of variability between groups compared to the variability within the groups determining whether sample means are within sampling variability of each other.

The larger F–measure depicts that the probability of real effects is statistically significant. When the effects are statistically significant, the means needs to be examined to determine the nature of the effects using Post-hoc test. The results presented in the Table 2, indicated that the value of ACI (MACI at temporal scale 1) for normal subjects is lower than CHF subject and higher than AF subjects resulting in incorrect dynamical information. When scale is increased, the MACI value for normal subjects became smaller than both CHF and AF subjects, showing that ACI is higher for both disease groups. The results also indicated that F-measure started to increase when temporal scales were increased. The increase in F-measure with temporal scales depicted that probability of real effects in separating healthy and pathological has increased. Since the results are statistically significant, the means needs to be examined to determine the nature of the effects using Post-hoc test.

In Table 3, Bonferroni post-hoc test is carried out for multiple comparisons of NSR and CHF, AF (disease) subjects. The results indicated that maximum separation between NSR and CHF subjects was obtained at temporal scale $\tau = 3$, (mean difference -0.1586 and p-value = $1.58 \times 10^{-18}$). Similarly, the maximum separation between NSR and AF subjects was

**Table 2. ANOVA table of ACI (MACI at temporal scale 1) and MACI (mean) indices for distinguishing NSR, CHF and AF subject.**

| Time Scale | Mean ± STD | | | Sum of Squares | | Df | | Mean Square | | F | Sig. |
|---|---|---|---|---|---|---|---|---|---|---|---|
| | NSR | CHF | AF | B/W Group | Within Group | B/W Group | Within Group | B/W Group | Within Group | | |
| 1 | 0.57±0.09 | 0.61±0.08 | 0.54±0.11 | 0.105 | 1.086 | 2 | 137 | 0.052 | 0.008 | 6.616 | $1.81 \times 10^{-3}$ |
| 2 | 0.52±0.09 | 0.63±0.07 | 0.64±0.10 | 0.500 | 1.039 | 2 | 137 | 0.250 | 0.008 | 32.96 | $2.06 \times 10^{-12}$ |
| 3 | 0.41±0.07 | 0.57±0.08 | 0.58±0.09 | 0.936 | 0.872 | 2 | 137 | 0.468 | 0.006 | 73.56 | $1.99 \times 10^{-22}$ |
| 4 | 0.42±0.06 | 0.53±0.09 | 0.56±0.07 | 0.561 | 0.738 | 2 | 137 | 0.280 | 0.005 | 52.05 | $1.53 \times 10^{-17}$ |
| 5 | 0.47±0.06 | 0.52±0.09 | 0.56±0.06 | 0.181 | 0.701 | 2 | 137 | 0.090 | 0.005 | 17.66 | $1.50 \times 10^{-7}$ |
| 6 | 0.50±0.06 | 0.50±0.07 | 0.57±0.05 | 0.091 | 0.541 | 2 | 137 | 0.046 | 0.004 | 11.57 | $2.27 \times 10^{-5}$ |
| 7 | 0.52±0.05 | 0.50±0.07 | 0.57±0.05 | 0.061 | 0.480 | 2 | 137 | 0.030 | 0.004 | 8.644 | $2.91 \times 10^{-4}$ |
| 8 | 0.53±0.04 | 0.50±0.07 | 0.56±0.05 | 0.057 | 0.404 | 2 | 137 | 0.029 | 0.003 | 9.743 | $1.10 \times 10^{-4}$ |
| 9 | 0.53±0.04 | 0.50±0.06 | 0.56±0.05 | 0.060 | 0.342 | 2 | 137 | 0.030 | 0.002 | 11.93 | $1.67 \times 10^{-5}$ |
| 10 | 0.53±0.04 | 0.49±0.06 | 0.57±0.06 | 0.095 | 0.339 | 2 | 137 | 0.047 | 0.002 | 19.14 | $4.67 \times 10^{-8}$ |

**Table 3. Comparisons of NSR, CHF and AF subjects on the basis of ACI (MACI at temporal scale 1) and MACI (mean).**

| Time Scale | Mean Difference of the groups | | | P-Value | | |
|---|---|---|---|---|---|---|
| | NSR VS CHF | NSR VS AF | CHF VS AF | NSR VS CHF | NSR VS AF | CHF VS AF |
| 1 | -0.0476 | 0.0284 | 0.0760 | $1.78 \times 10^{-2}$ | $5.34 \times 10^{-1}$ | $2.97 \times 10^{-3}$ |
| 2 | -0.1145 | -0.1278 | -0.0133 | $6.07 \times 10^{-10}$ | $1.62 \times 10^{-8}$ | 1 |
| 3 | -0.1586 | -0.1721 | -0.0135 | $1.58 \times 10^{-18}$ | $2.07 \times 10^{-15}$ | 1 |
| 4 | -0.1141 | -0.1448 | -0.0306 | $6.93 \times 10^{-13}$ | $1.78 \times 10^{-13}$ | $3.06 \times 10^{-1}$ |
| 5 | -0.0538 | -0.0919 | -0.0381 | $3.99 \times 10^{-4}$ | $6.79 \times 10^{-7}$ | $1.13 \times 10^{-1}$ |
| 6 | -0.0028 | -0.0688 | -0.0660 | 1 | $2.38 \times 10^{-5}$ | $1.82 \times 10^{-4}$ |
| 7 | 0.0202 | -0.0422 | -0.0624 | $2.30 \times 10^{-1}$ | $8.94 \times 10^{-3}$ | $1.71 \times 10^{-4}$ |
| 8 | 0.0309 | -0.0280 | -0.0589 | $1.04 \times 10^{-2}$ | $9.14 \times 10^{-2}$ | $1.07 \times 10^{-4}$ |
| 9 | 0.0288 | -0.0321 | -0.0610 | $9.19 \times 10^{-3}$ | $2.16 \times 10^{-2}$ | $1.19 \times 10^{-5}$ |
| 10 | 0.0347 | -0.0426 | -0.0773 | $1.12 \times 10^{-3}$ | $1.19 \times 10^{-3}$ | $2.73 \times 10^{-8}$ |

obtained at temporal scale τ = 3 (mean difference -0.1721, sig. = $2.07 \times 10^{-15}$) and CHF and AF subjects was obtained at temporal scale τ = 10 (mean difference -0.0773, p-value = $2.73 \times 10^{-8}$).

It is evident from Table 2, for temporal scales 2 and 3, mean MACI values of CHF and AF pathological groups are almost same, however, interbeat interval time series of both pathological groups are outcome of different cardiac dynamics. Similarly, at temporal scales 6 and 7, mean MACI of NSR and CHF subjects almost same. Thus, to differentiate the time series of dynamical processes, both specific values of MACI and the dependence of MACI values at temporal scales needs to take into consideration.

In Fig 3, comparison ACI and MACI (optimal scale) for separation of NSR vs CHF, NSR Vs AF and NSR Vs CHF are shown. It is evident from the figure, ACI was modest in separating these groups, however, MACI at optimal threshold values revealed significantly high

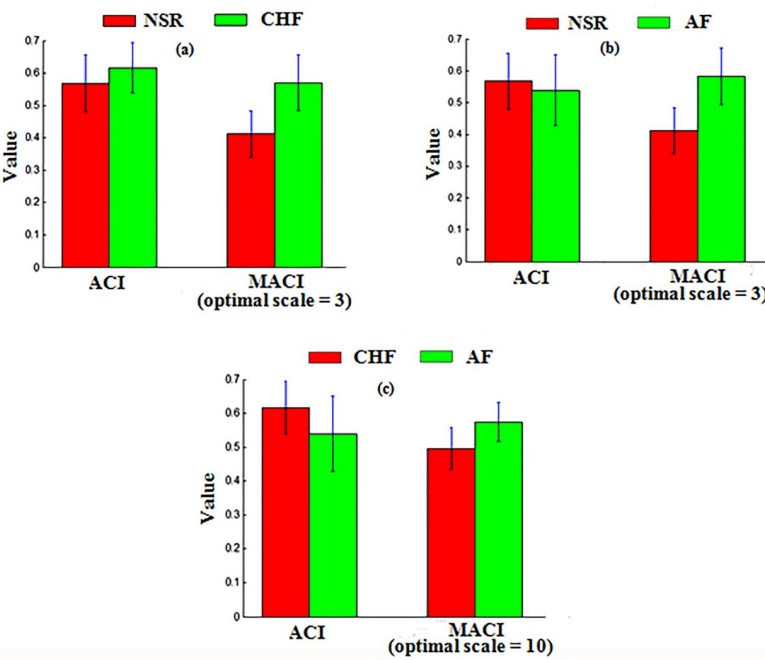

**Fig 3.** Comparison of ACI and MACI (at optimal scale) for classification of (a) NSR Vs CHF, (b) NSR Vs AF and (c) CHF Vs AF subjects.

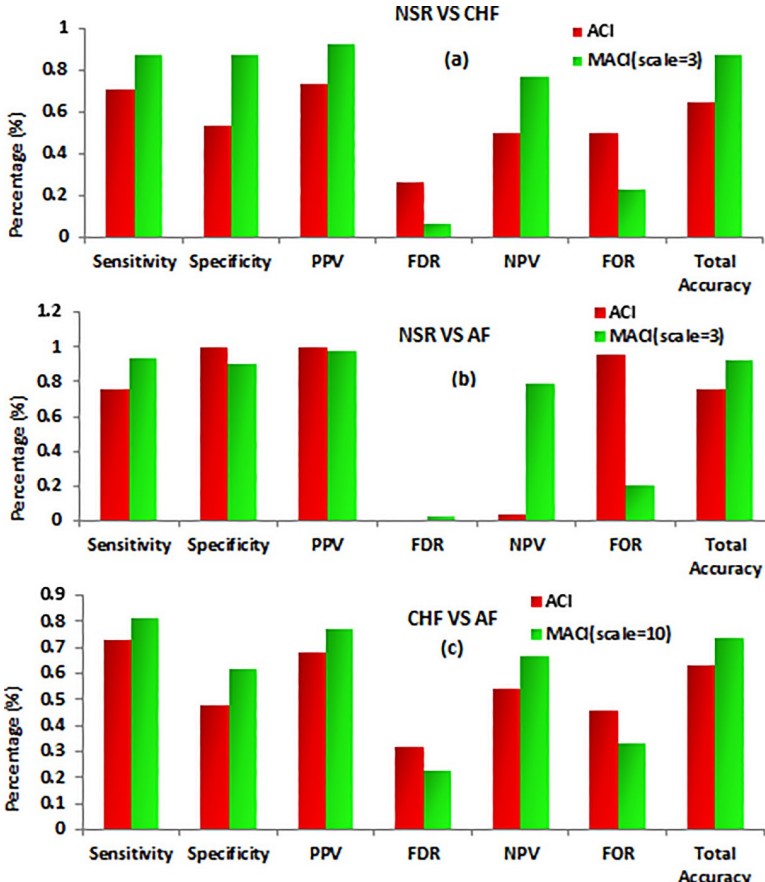

**Fig 4.** Performance evaluation of ACI and MACI (at optimal scale value) for classification of (a) NSR Vs CHF, (b) NSR Vs AF and (c) CHF Vs AF subjects.

separation between the different groups. MACI showed highest separation between NSR and both pathological groups (CHF and AF) at optimal scale 3 and highest separation between two pathological groups CHF Vs AF was obtained at temporal scale 10. The findings aver that small temporal scale can distinguish healthy and pathological groups due to considerable difference in the dynamical information provided by HRV signals, however, for separating path.

In Fig 4, performance evaluation metrics sensitivity, specificity, positive predictive value (PPV), false discovery rate (FDR), negative predictive value (NPV), false omission rate (FOR) and total accuracy are shown for classification of NSR vs CHF, NSR Vs AF and NSR Vs CHF subjects using the Multilayer perceptron and 10-fold cross validation strategy. It is evident from figure all the evaluation metrics revealed higher classification ability of MACI at optimal scales for classification of different groups.

In Fig 5, the area under ROC curve (AUC) is shown for assessing the degree of separation between NSR Vs CHF, NSR Vs AF and CHF and AF subjects by plotting "1-specificity" against "sensitivity". AUC is a well-established index of diagnostic accuracy. The AUC = 0.5 shows picking a class by a pure chance and AUC = 1 reveals perfect separation between two classes. Higher values of AUC for MACI reveal higher classification ability of this index at optimal time scales compared ACI.

In Fig 6, results of MACI for different signal lengths to classify NSR and CHF subjects are illustrated. The findings show that optimal separation between NSR and CHF was found at

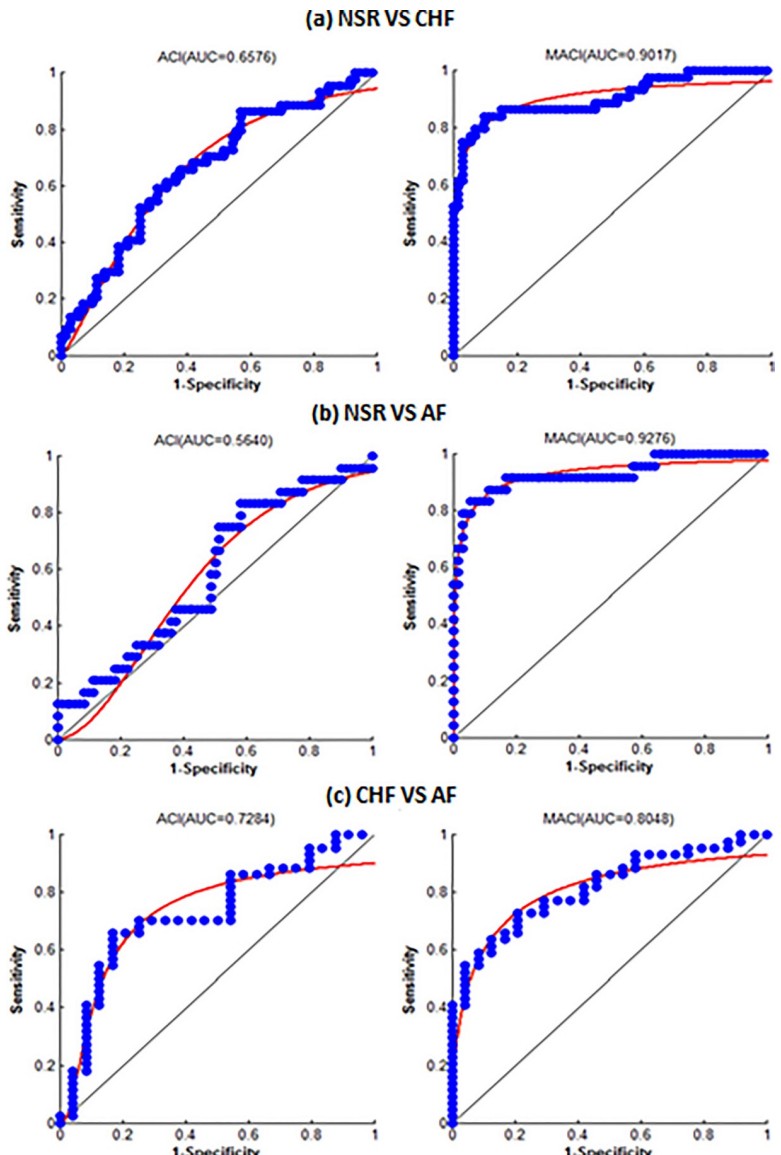

**Fig 5.** AUC for assessing degree of separation using ACI and MACI at optimal scale (a) NSR Vs CHF (b) NSR Vs AF and (c) CHF Vs AF subjects.

time scale 3 for signal length >1000, and for signal lengths < = 1000, time scale 2 provided maximum separation followed by the time scale 3. The results demonstrate that almost for all the signal lengths optimal time scale 3 is consistent for classifying NSR and CHF subjects.

In Fig 7, switching averaging brackets to right to investigate that coarse graining procedure is stable or not. We investigated the effect by switching the averaging brackets 1 to 4 steps to right. WE found overlap of MACI values at switching brackets 1 to 4 steps. It is inferred out for small switching brackets coarse graining procedure is stable and does not affect the MACI at different threshold values.

Furthermore, the performance of MACI was evaluated by comparing it with other nonlinear multiscale based techniques (Improved Multiscale Permutation Entropy (IMPE) [21] and Multiscale Normalized Corrected Shannon Entropy (MNCSE) [22]). The effectiveness of the

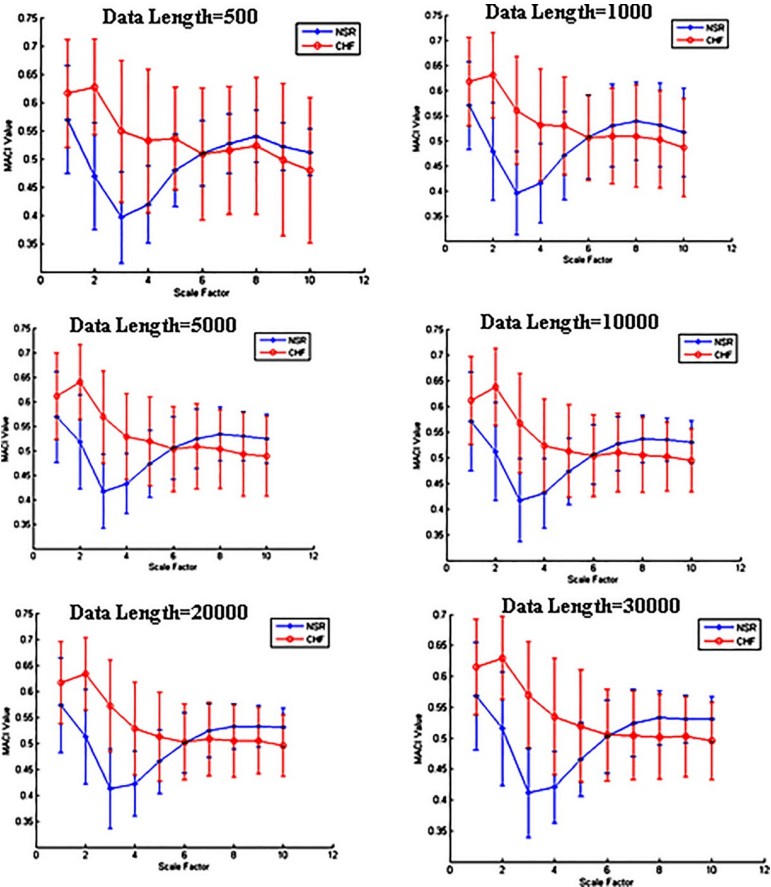

**Fig 6. Variations in the MACI values at different time scales with signal length.**

proposed approach is demonstrated using interbeat interval signals from healthy and pathological subjects. The only column labelled Sig., representing significance level and if the value (s) of Sig. in this column is (are) less than the critical value set by the researcher (say, 0.05), then the outcome will result in significant effects, while value(s) greater 0.05 value will depict that effects are not statistically significant. The statistically significant results demonstrate that differences between groups are not by chance or sampling error.

In Table 4, corresponding p-values of MACI, IMPE and MNCSE for healthy and pathological subjects are presented at temporal scales 1 to 10. Scales based measure MACI, IMPE and MNCSE were able to discriminate healthy from pathological group more significantly at a wide range of scales. The maximum separation between healthy and pathological subjects was obtained at temporal scale 3(p-value = $1.58 \times 10^{-18}$) for MACI, temporal scale 7(p-value = $1.00 \times 10^{-2}$) for IMPE and temporal scale 4(p-value = $2.72 \times 10^{-6}$) for MNCSE. It is evident that the differences between MACI, IMPE and MNCSE estimates were smaller at scale 1, as compared to multiple time scale. The MACI provided was more robust in distinguishing healthy and pathological subjects.

## Discussion

The cardiac autonomic control is regulated by the interacting mechanism of sympathetic and parasympathetic branches of autonomic nervous system (ANS) and its ability to adapt and

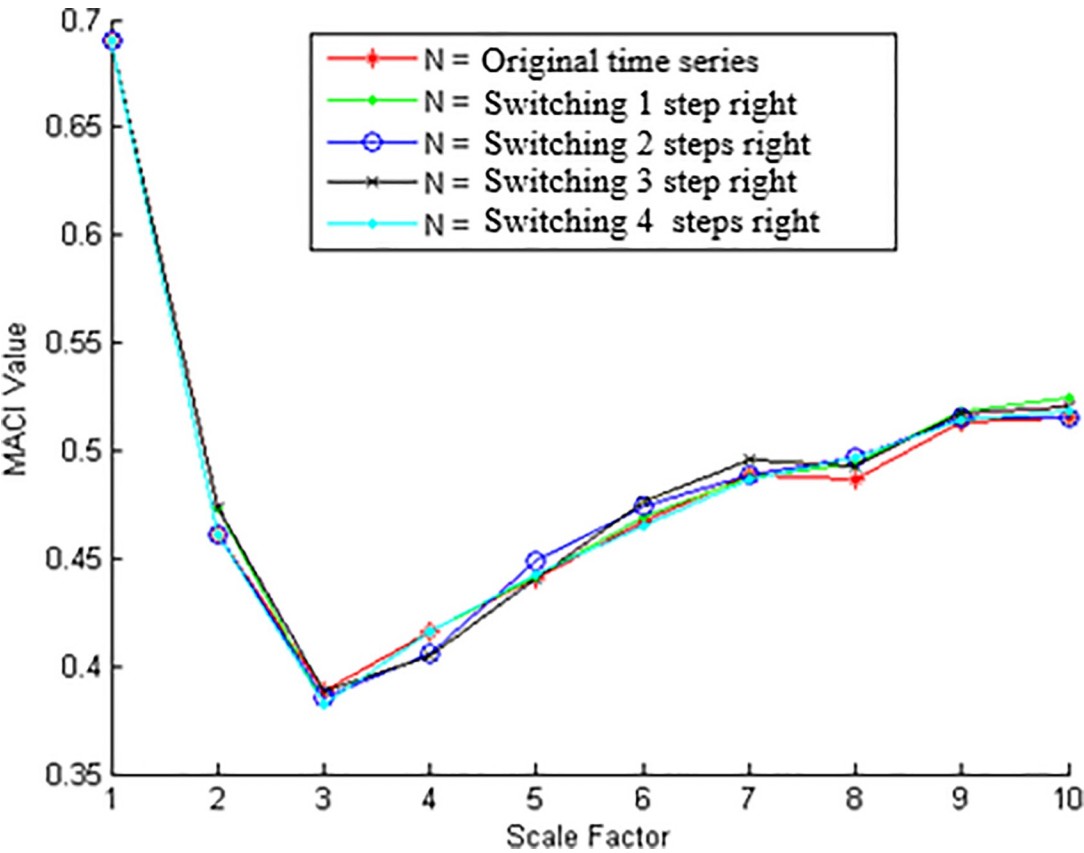

**Fig 7. Switching of averaging brackets to the right.**

function in a dynamic environment. Higher adaptive of capability of ANS to regulate cardiac autonomic control reveals HRV signals of heathy subject reflect higher complexity [2] and decreased ACI [13, 16]. Aging and disease reduce the adaptive capability of the ANS and degrade cardiac autonomic, which results in loss of complexity and increase in ACI. The recent research evidences [2, 21, 22] reveal that studying dynamics of these signals at traditional single scale provides misleading information. In this study, we analysed the dynamics of

**Table 4. Corresponding p-values comparison of MACI, IMPE and MNCSE at temporal scales 1 to 10 for quantifying the dynamics of healthy (NSR) and pathological (CHF) subjects.**

| Time Scale | Sig. | | |
|---|---|---|---|
| | **MACI** | **IMPE** | **MNCSE** |
| 1 | $1.78 \times 10^{-2}$ | $5.10 \times 10^{-1}$ | $6.62 \times 10^{-1}$ |
| 2 | $6.07 \times 10^{-10}$ | $3.46 \times 10^{-1}$ | $2.18 \times 10^{-3}$ |
| 3 | $1.58 \times 10^{-18}$ | $8.51 \times 10^{-1}$ | $6.26 \times 10^{-5}$ |
| 4 | $6.93 \times 10^{-13}$ | $2.86 \times 10^{-1}$ | $2.72 \times 10^{-6}$ |
| 5 | $3.99 \times 10^{-4}$ | $1.16 \times 10^{-1}$ | $2.72 \times 10^{-6}$ |
| 6 | 1 | $1.70 \times 10^{-2}$ | $2.94 \times 10^{-6}$ |
| 7 | $2.30 \times 10^{-1}$ | $1.00 \times 10^{-2}$ | $8.98 \times 10^{-6}$ |
| 8 | $1.04 \times 10^{-2}$ | $7.60 \times 10^{-3}$ | $1.12 \times 10^{-5}$ |
| 9 | $9.19 \times 10^{-3}$ | $1.10 \times 10^{-2}$ | $2.11 \times 10^{-5}$ |
| 10 | $1.12 \times 10^{-3}$ | $1.10 \times 10^{-2}$ | $5.13 \times 10^{-5}$ |

healthy and pathological using a novel index MACI to investigative robustness of incorporation of multiple time to quantify dynamics of HRV signals extracted from ECG recording of NSR, CHF and AF subjects.

It is evident from Table 2 and Fig 3, ACI (TACI at $\tau = 1$) showed dynamically incorrect results for AF subjects (0.54±0.11) compared to NSR (0.57±0.09). Incorporating multiple time scales resulted in lower ACI for NSR subjects compared to both AF and CHF subjects revealing that dynamical route of disease is associated with increase of ACI. Basically, ACI increases when local maxima are immediately followed by local minima and vice versa [13, 16]. In healthy subjects controlling mechanism of ANS regulates heartbeat and controls the immediate alternations of local maxima and local minima, which results is decrease in ACI in healthy subjects. However, in case pathological subjects due to loss of adaptive capability, controlling mechanism of ANS fails to regulate heartbeat, which causes immediate alternations in heartbeat, which results in the increase of ACI. Thus, increase in ACI is associated with the dynamics route of disease and decrease in ACI is associated with healthy dynamics.

We used standard multiscaling procedure proposed by Costa et al [2] for generating coarse grained time series. We observed significantly lower MACI for NSR subjects compared to CHF and AF subjects at time scales 2 to 5 and at time scales 2 to 10 respectively. The significant difference between CHF and AF was observed at times scales 4 to 10. The highest separation between NSR vs CHF and NSR vs AF was obtained at time scale 3, whereas for NSR vs AF subjects, maximum separation was obtained at time scale 10. Thus, we merely iterated time scale iterated time scale values from 1 to 10 to characterize the heart rate dynamics of healthy and pathological subjects. The findings aver that along with specific MACI values, we need to consider their dependence on time scale for better characterization of HRV signals of healthy and pathological subjects.

The present study investigated dynamics of heart rate variability signals to assess cardiac autonomic control of nervous system. Thus, we need not to consider the morphology of the ECG signal, instead, we extracted RR-intervals (interbeat intervals) from the ECG signal and used MACI, which is an extension ACI for assessing dynamical fluctuations of healthy and different pathological subjects. The sign change in the time series occurs when an increasing interval immediately is followed by the decreasing interval and vice versa. Thus, sign information is enough for investigating the dynamical fluctuations of the heart, which assist the clinicians to asses cardiac autonomic control and its variations occurring due to aging and disease.

We also investigated the variations in MACI values for different signal lengths to classify NSR and CHF subjects. The findings show that optimal separation between NSR and CHF was found at time scale 3 for signal length >1000, and for signal lengths < = 1000, time scale 2 provided maximum separation followed by the time scale 3. The results demonstrate that almost for all the signal lengths optimal time scale 3 is consistent for classifying NSR and CHF subjects. The optimal value of time scale revealing highest separation between NSR vs CHF and NSR vs AF was 3, whereas for NSR vs AF subjects optimal scale was 10. Thus, optimal scale is different for separating NSR subject from pathological (NSR Vs CHF of NSR Vs AF) subjects compared to optimal scale for separating two pathological groups (CHF Vs AF). The findings aver that along with specific MACI values, we need to consider their dependence on time scale for better characterization of HRV signals of healthy and pathological subjects.

## Conclusion

In this study, we proposed multiscale acceleration change (MACI) to analyse the dynamics of HRV signals accurately and improve classification ability of ACI. to classify healthy and pathological subjects. The preliminary results aver that MACI provided dynamically more accurate

information compared ACI and outperformed ACI at wide range of temporal scales in classi-fying healthy and pathological subjects. We also compare MACI with IMPE and MNCSE for assessing the robustness the proposed index. The performance parameters clearly demon-strated that proposed MACI provided better separation between these groups than traditional ACI, IMPE and MNCSE measures. The contributions of this research are twofold. The empiri-cal results show that classification ability of ACI has increased by incorporation of multiple time scales and this index more effective in analysing dynamics HRV signals accurately. The present investigated performance of MACI for classifying only HRV signals of healthy and pathological subjects. Further studies are suggested to evaluate the robustness of MACI for analysing other biological signals such as electroencephalographic recording and stride inter-val time series of healthy and neurodegenerative disease subjects.

## Author Contributions

**Conceptualization:** Syed Zaki Hassan Kazmi, Syed Ali Abbas.

**Data curation:** Nazneen Habib.

**Formal analysis:** Syed Zaki Hassan Kazmi, Rabia Riaz.

**Funding acquisition:** Tae-Sun Chung.

**Investigation:** Syed Zaki Hassan Kazmi, Rabia Riaz.

**Methodology:** Nazneen Habib, Syed Ali Abbas.

**Project administration:** Rabia Riaz, Sanam Shahla Rizvi, Tae-Sun Chung.

**Resources:** Nazneen Habib, Rabia Riaz, Sanam Shahla Rizvi, Tae-Sun Chung.

**Software:** Sanam Shahla Rizvi.

**Supervision:** Sanam Shahla Rizvi, Tae-Sun Chung.

**Validation:** Rabia Riaz, Sanam Shahla Rizvi, Syed Ali Abbas, Tae-Sun Chung.

**Visualization:** Syed Zaki Hassan Kazmi.

**Writing – original draft:** Syed Ali Abbas.

**Writing – review & editing:** Nazneen Habib, Sanam Shahla Rizvi.

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
