## [Decision Letter · Decision Letter 0]

20 Jul 2020

PONE-D-20-07743

MULTISCALE BASED NONLINEAR DYNAMICS ANALYSIS OF HEART RATE VARIABILITY SIGNALS

PLOS ONE

Dear Dr. Kwon,

Thank you for submitting your manuscript to PLOS ONE. After careful consideration, we feel that it has merit but does not fully meet PLOS ONE’s publication criteria as it currently stands. Therefore, we invite you to submit a revised version of the manuscript that addresses the points raised during the review process.

Please address all comments indicated by the Reviewers.

We look forward to receiving your revised manuscript.

Kind regards,

Elena G. Tolkacheva, PhD

Academic Editor

PLOS ONE

Journal Requirements:

2. Thank you for inlcuding your competing interests statement; "The authors have declared that no competing interests exist."

We note that one or more of the authors are employed by a commercial company:

Raptor Interactive (Pty) Ltd

Please include both an updated Funding Statement and Competing Interests Statement in your cover letter. We will change the online submission form on your behalf

Reviewers' comments:

Reviewer's Responses to Questions

**Comments to the Author**

1. Is the manuscript technically sound, and do the data support the conclusions?

Reviewer #1: No

Reviewer #2: Partly

Reviewer #3: Partly

2. Has the statistical analysis been performed appropriately and rigorously? 

Reviewer #1: No

Reviewer #2: Yes

Reviewer #3: No

3. Have the authors made all data underlying the findings in their manuscript fully available?

Reviewer #1: Yes

Reviewer #2: Yes

Reviewer #3: Yes

4. Is the manuscript presented in an intelligible fashion and written in standard English?

Reviewer #1: Yes

Reviewer #2: Yes

Reviewer #3: Yes

5. Review Comments to the Author

Reviewer #1: This paper discusses a multiscale based nonlinear dynamics analysis of heart rate variability signals. The presentation of the content is fine and the data is well explained. However, it is unclear what is new here compared with the existing methods. In addition, the significance of the results is lacked. Therefore, I suggest to decline this paper for publication.

Reviewer #2: The proposed method of using time evolution of heart rate variability captured by the so called Multi-scale based Acceleration Change Index for heart abnormality detection is very interesting. I have the following suggestions before the publication.

The abstract is too long. I would suggest to make the abstract concise and remove the . For instance, the importance of heart rhythm monitoring for early diagnosis of disease using ECG signals is a well-known fact and that part can be summarize.

I suggest you add a more comprehensive References. For instance there exist recent works on predictive analysis of ECG signals that solve similar problems.

Use consistent format for acronyms. For instance lowercase first letter is used for "differentiated time series (DNN)" while uppercase first letter is used for "Acceleration Change Index (ACI)".

Please clarify why the sign information (the slope of acceleration) is sufficient for some diagnosis cases, as it is known that there are many abnormalities like "supraventicular arrhythmia" that can be captured by analyzing the morphology of the ECG signal.

The results show a strong improvement for MACI vs ACI analysis in terms of the area under the ROC curve.

I would encourage you compare your method to some existing methods in addition to the internal comparison between the ACI and MCI approaches.

Reviewer #3: Based on an existing index Acceleration Change Index (ACI) for describing heart rate variability (HRV), this paper proposes to feed coarse-grained HRV data to ACI. The so-called multiscale ACI (MACI) seems to improve the performance of ACI in distinguishing between healthy and pathologic subjects. However, more validations are needed. Moreover, comparison to other methods is missing despite so many existing methods for this purpose.

Specific comments:

1. The proposed approach is called multiscale ACI. However, in the end, only one scale is used. The current approach is more like an ACI on coarse-grained time series with a tunable scale parameter. Can you use different scales at the same time to make this a real multiscale approach? A simple example would be just taking the average over a few $\\tau$ values. Of course, linear or non-linear models can be trained using multiple scales but this will involve more validation such as training-testing splits.

2. Is the coarse-graining procedure stable? In other words, if the averaging brackets are switched one step to the right (e.g., deleting x1 in the time series), will the new index change a lot? A more detailed quantitative justification for this would be helpful.

3. This is related to comment 1. In Fig. 2, how is the choice of whether to use more scale made? Are you merely iterating over a predefined set of $\\tau$ values? Or is there any criteria to decide whether to try more $\\tau$ values given the current computation results?

4. In the applications, there is an optimal $\\tau$ value of 3. It would be helpful to test whether this optimal value is consistent, e.g., by subsampling the datasets and see if 3 is still an optimal value.

5. The performance advantage of the coarse-grained approach is clear, but there is little interpretation presented. Can you use some example time series and visualize the difference among ACI, MACI with optimal $\\tau$, MACI with suboptimal $\\tau$ on the intermediate calculation results as described in the ACI algorithm.

6. Due to the small data size, I recommend using some statistical tests to further support the performance improvements observed in Fig. 4 and 7.

7. There are so many other indices for characterizing HRV, such as the ones listed in “Shaffer, Fred, and J. P. Ginsberg. "An overview of heart rate variability metrics and norms." Frontiers in public health 5 (2017): 258.” Comparison to the popular ones should be included. The paper that introduced ACI (Gonzalez et al 2003) did do some comparison to four other indices. But that comparison was on a different dataset and more methods have been proposed during the pase 17 years. Therefore, a fair comparison to other methods on the dataset used in this paper should be included.

8. This is related to comment 7. In the first paragraph of “Materials and Methods”, the authors claimed that “no other index except ACI provide significant difference between the groups”. In the original publication which this paper heavily based on, the authors compared ACI with four other indices and found that ACI is the best among the five. Thus, this statement in its current form is too strong and the context of Gonzalez et al (2003) should be included here.

9. There are many grammar errors in the manuscript. I strongly recommend the authors to carefully proofread the manuscript.

6. PLOS authors have the option to publish the peer review history of their article (what does this mean?). If published, this will include your full peer review and any attached files.

Reviewer #1: No

Reviewer #2: **Yes: **Dr. Abolfazl Razi

Reviewer #3: No

---

## [Author Response · Author response to Decision Letter 0]

21 Oct 2020

Kindly see the attached Response to Reviewers file.

---

## [Decision Letter · Decision Letter 1]

23 Nov 2020

MULTISCALE BASED NONLINEAR DYNAMICS ANALYSIS OF HEART RATE VARIABILITY SIGNALS

PONE-D-20-07743R1

Dear Dr. Chung,

We’re pleased to inform you that your manuscript has been judged scientifically suitable for publication and will be formally accepted for publication once it meets all outstanding technical requirements.

Kind regards,

Elena G. Tolkacheva, PhD

Academic Editor

PLOS ONE

Additional Editor Comments (optional):

Reviewers' comments:

Reviewer's Responses to Questions

**Comments to the Author**

1. If the authors have adequately addressed your comments raised in a previous round of review and you feel that this manuscript is now acceptable for publication, you may indicate that here to bypass the “Comments to the Author” section, enter your conflict of interest statement in the “Confidential to Editor” section, and submit your "Accept" recommendation.

Reviewer #1: All comments have been addressed

Reviewer #3: All comments have been addressed

2. Is the manuscript technically sound, and do the data support the conclusions?

Reviewer #1: Yes

Reviewer #3: Partly

3. Has the statistical analysis been performed appropriately and rigorously? 

Reviewer #1: Yes

Reviewer #3: Yes

4. Have the authors made all data underlying the findings in their manuscript fully available?

Reviewer #1: Yes

Reviewer #3: Yes

5. Is the manuscript presented in an intelligible fashion and written in standard English?

Reviewer #1: Yes

Reviewer #3: Yes

6. Review Comments to the Author

Reviewer #1: The authors made a substantial change, which makes the revision much more clear than the original submission. I think this version is ready for publication.

Reviewer #3: (No Response)

7. PLOS authors have the option to publish the peer review history of their article (what does this mean?). If published, this will include your full peer review and any attached files.

Reviewer #1: No

Reviewer #3: No

---

## [Editor Report · Acceptance letter]

2 Dec 2020

PONE-D-20-07743R1 

Multiscale based nonlinear dynamics analysis of heart rate variability signals 

Dear Dr. Chung:

I'm pleased to inform you that your manuscript has been deemed suitable for publication in PLOS ONE. Congratulations! Your manuscript is now with our production department. 

Kind regards, 

on behalf of

Dr. Elena G. Tolkacheva 

Academic Editor

PLOS ONE